# Extended Runge-Kutta Scheme and Neural Network Approach for SEIR Epidemic Model with Convex Incidence Rate

**Ahmed A. Al Ghafli** [1],*[ID]**, Yasir Nawaz** [2],*[ID]**, Hassan J. Al Salman** [1][ID] **and Muavia Mansoor** [3]

1   Department of Mathematics and Statistics, College of Science, King Faisal University,
    Hofuf 31982, Al-Ahsa, Saudi Arabia; hjhalsalman@kfu.edu.sa
2   Department of Mathematics, Air University, PAF Complex E-9, Islamabad 44000, Pakistan
3   Department of Mathematics, University of Wah, Wah Cantt 47040, Pakistan; muavia.mansoor@uow.edu.pk
*   Correspondence: aamgkafli@kfu.edu.sa (A.A.A.G.); yasir_maths@yahoo.com (Y.N.)

**Abstract:** For solving first-order linear and nonlinear differential equations, a new two-stage implicit–explicit approach is given. The scheme's first stage, or predictor stage, is implicit, while the scheme's second stage is explicit. The first stage of the proposed scheme is an extended form of the existing Runge–Kutta scheme. The scheme's stability and consistency are also offered. In two phases, the technique achieves third-order accuracy. The method is applied to the SEIR epidemic model with a convex incidence rate. The local stability is also examined. The technique is evaluated compared to existing Euler and nonstandard finite difference methods. In terms of accuracy, the produced plots show that the suggested scheme outperforms the existing Euler and nonstandard finite difference methods. Furthermore, a neural network technique is being considered to map the relationship between time and the amount of susceptible, exposed, and infected people.

**Keywords:** two-stage scheme; stability; consistency; neural network; SEIR model

## 1. Introduction

The models of epidemics are of great interest to mathematicians, including those [1] that deal with the COVID-19 epidemic model that considers the effects of lock-down and the possibility of the transmission of the disease from dead bodies to humans. It is mentioned in [2] that the COVID-19 pandemic was the most significant global crisis after the Second World War. The interaction of the people and the infections reservoir (seafood market) and the interaction of bats and unknown hosts has been given [3]. A numerical scheme based on the Adams–Bashforth and the Fourier spectral methods has been formulated to solve Belousov–Zhabotinskii reaction systems [4]. These schemes can be used to solve any model of epidemic disease containing space and time derivative terms. The smoking model and its public health have been studied [5]. The study also contained the existence and uniqueness problems via fixed point theory. Epidemiology research is crucial for understanding disease prevalence in the general population. Building models, estimating parameters, determining how sensitive models are to parameter changes, and doing numerical simulations are frequently the tasks involved in mathematical epidemiology. To better understand past illness outbreaks and the dynamics of how infections propagate through populations, epidemiologists employ mathematical models [6]. This type of research aids in understanding the population's illness spread ratio and managing its characteristics [7,8]. These forms of disease models are frequently referred to as contagious diseases or diseases that spread from one person to another. To solve a set of differential equations that explain the behavior of infectious illnesses in a population, mathematical procedures known as numerical schemes are used.

Epidemiologists utilize these methods to model the spread of illnesses and learn more about the dynamics of epidemics. Since systems of nonlinear differential equations that make up epidemic models are frequently difficult to solve explicitly, numerical approaches

such as [9–11] are used to approximate the behavior of the underlying system. To analyze epidemic models numerically, compartmental models, spatial models, and network-based models have all been established. These methods enable researchers to investigate the effects of various treatments on the spread of a disease, such as vaccination, quarantine, and social isolation. Public health officials may also create efficient plans for reducing the effects of pandemics and make well-informed judgments about how to restrict the spread of a disease using numerical systems for epidemic models [12,13].

The susceptible, exposed, infectious, and recovered (SEIR) [14,15] model is a mathematical simulation of how infectious diseases move through a community. Susceptible, exposed, infectious, and recovered are the four distinct stages of the infection cycle upon which it is built. The dynamics of infectious disease outbreaks can be accurately predicted using this model, which is frequently employed in epidemiology research. Understanding how various treatments, such as immunizations or social isolation practices, can affect the transmission of infectious diseases is made easier by using the SEIR model. The SEIR model aids researchers and public health professionals in making well-informed decisions about how to manage disease outbreaks best and safeguard public health by taking into consideration aspects like transmission rates and population demographics. Y. Qureshi et al. [16] used the AB fractional operator to study the dynamics of diarrhea. They discovered that when the AB fractional order operator, which has a non-local and non-singular kernel, is considered, the investigated diarrhea model adequately approximates the real statistical data. By taking into account the AB fractional derivative, Bas and Ozarslan [17] looked into the analytical solutions of a few dynamical models, including the logistic equation, population growth/decay equation, and the blood alcohol model. This model was proposed in 1927 by Kermack and McKendrick [18,19]. The spatio-temporal epidemic model has been studied in [20], and particular attention has been paid to the hair-trigger effect. The assumptions of random recovery or death of ill individuals and constant infectivity were dropped [21] in the existing model of a general epidemic. A deterministic model has been studied [22] for the spatial spread of an epidemic. The solution of the model had a temporally asymptotic limit. The Kermack and McKendrick model was reformulated in [23] as a nonlinear age-dependent population dynamics model. Some studies of the Kermack and McKendrick mathematical models can be seen in [24].

Numerous numerical techniques, for example, finite difference, finite element, and finite volume, are available in the literature for solving science and engineering problems. There are two types of finite difference schemes, categorized as explicit and implicit. Explicit schemes may converge quicker than implicit schemes, and they may not necessitate the linearized form of differential equations. Generally, implicit schemes may provide a larger stability region than explicit methods. As a result, big step sizes can be advantageous in solving problems when employing implicit schemes. Some implicit schemes are unconditionally stable, allowing very high step sizes, although big step sizes increase the solution's error. This paper proposes an implicit-explicit approach for solving linear and nonlinear differential equations, but it is different from most of the existing schemes because it contains the addition of an extra second-order derivative term in the first implicit stage of the scheme. Including the second-order term enhances the accuracy of the solution in two stages. The scheme can be applied to any initial and boundary value problem, but for this work, it is applied to an epidemic model. Since the solution of an epidemic stays positive for some choice of parameters, so can any high-order scheme be employed in those situations to get a more accurate solution. For initial value problems, the suggested scheme can be combined with any iterative scheme; however, for second- or higher-order boundary value problems, another strategy known as the shooting method can be used with the proposed scheme. Thus, the proposed, iterative, and shooting methods can simultaneously solve second- or higher-order boundary value problems. In this way, one can solve many differential equations.

The paper is organized as follows: Section 2 is concerned with constructing a numerical scheme with stability and consistency analysis, Section 3 deals with a mathematical

model and local stability analysis, Section 4 consists of the results and discussion, and Section 5 concludes.

## 2. Numerical Scheme

The proposed scheme is an implicit-explicit scheme with a third-order of accuracy. The scheme can be considered an extended form of the Runge–Kutta scheme. The disadvantage of the scheme is the need to calculate an extra derivative, but it has the advantage of providing third-order accuracy in two stages. It consists of two grid points or two time levels. Because the scheme is a predictor–corrector scheme, so does the second or corrector stage use the information of the first or predictor stage. The scheme can be applied to first-order linear and nonlinear differential equations. The application of the scheme is given for a system of first-order nonlinear ordinary differential equations. For applying it to second- or third-order boundary value problems, it can be applied with a combination of shooting methods.

For proposing a numerical scheme to solve differential equations, consider a nonlinear first-order differential equation,

$$y' = f(y),\tag{1}$$

subject to the initial condition,

$$f(y) = \beta,\tag{2}$$

where $\beta$ is a constant.

For solving Equations (1) and (2) using the proposed scheme, the first stage is given as,

$$\overline{y}_{i+1} = y_i + chy'_{i+1} + h^2 y''_i\tag{3}$$

where $c$ any parameter to be is determined later. The solution $\overline{y}_{i+1}$ is a predicted one that will be corrected by the second stage of the scheme. The second-order derivative in Equation (3) can be found by applying a derivative to Equation (1) as,

$$y'' = f'(y)y'\tag{4}$$

The symbol "$h$" in Equation (3) is the step size.

The second stage of the scheme can be expressed as,

$$y_{i+1} = \frac{1}{4}\left(3y_i + \overline{y}_{i+1}\right) + h\left(ay'_i + b\overline{y}_{i+1}\right).\tag{5}$$

By looking at Equation (3), it can be observed that this stage of the scheme is implicit. The second stage (5) of the scheme is explicit. For finding unknowns $a$ and $b$ in Equation (5), the procedure is given as under.

In the beginning, substituting Equation (3) into Equation (5) as,

$$y_{i+1} = \frac{1}{4}\left(4y_i + chy'_{i+1} + h^2 y''_i\right) + h\left(ay'_i + by'_i + bchy''_{i+1} + bh^2 y'''_i\right).\tag{6}$$

Substituting Taylor series expansions for $y_{i+1}$, $y'_{i+1}$ and $y''_{i+1}$ into Equation (6) yields,

$$y_i + hy'_i + \frac{h^2}{2}y''_i + \frac{h^3}{6}y'''_i = \frac{1}{4}\left(4y_i + chy'_i + ch^2 y''_i + c\frac{h^3}{2}y'''_i + h^2 y''_i\right) + \\ h\left(ay'_i + by'_i + bchy''_i + bch^2 y'''_i + bh^2 y'''_i\right).\tag{7}$$

By equating $hy'_i$, $h^2 y''_i$ and $h^3 y'''_i$ on both sides of Equation (7) yields,

$$1 = \frac{c}{4} + a + b,\tag{8}$$

$$\frac{1}{2} = \frac{c}{4} + \frac{1}{4} + bc,\tag{9}$$

$$\frac{1}{6} = \frac{c}{8} + bc + b, \tag{10}$$

Since, Equations (8)–(10) are a nonlinear system of equations, a MATLAB solver *fsolve* is adopted to solve the equations. Using different initial guesses, two sets of solutions are obtained as,

$$a = 0.7470, \, b = 0.0288, \, c = 0.8968, \tag{11}$$

and

$$a = 1.9196, \, b = -0.3621, \, c = -2.2301. \tag{12}$$

Therefore for discretizing Equation (1) using the proposed scheme, two stages of the scheme are expressed as,

$$\overline{y}_{i+1} = y_i + 0.8968 h f(y_{i+1}) + h^2 f'(y_i) f(y_i), \tag{13}$$

$$y_{i+1} = \frac{1}{4}(3y_i + \overline{y}_{i+1}) + h(0.7470 f(y_i)) + 0.0288 f(\overline{y}_{i+1}). \tag{14}$$

or

$$\overline{y}_{i+1} = y_i - 2.2301 h f(y_{i+1}) + h^2 f'(y_i) f(y_i), \tag{15}$$

$$y_{i+1} = \frac{1}{4}(3y_i + \overline{y}_{i+1}) + h(1.9196 f(y_i)) - 0.3621 f(\overline{y}_{i+1}). \tag{16}$$

### 2.1. Stability Analysis

Here, the stability analysis is given for a system of first-order linear differential equations. However, if the system is nonlinear, it can be linearized, and the same procedure will be adopted for a linearized system of equations. Let the system of the linearized or linear differential equation be given as,

$$V' = AV, \tag{17}$$

where $V$ is a vector of dependent variables, and $A$ is a square matrix. For a linearized system, $A$ can be a Jacobean evaluated at the equilibrium point. The Equation (17) can be discretized by the proposed scheme, and the discretized equations are expressed as,

$$\overline{V}_{i+1} = V_i + chAV_{i+1} + h^2 A^2 V_i, \tag{18}$$

$$V_{i+1} = \frac{1}{4}(3V_i + \overline{V}_{i+1}) + h(aAV_i + bAV_{i+1}). \tag{19}$$

Since the proposed scheme is implicit–explicit, for solving difference equations obtained by applying the proposed scheme to Equation (17), a Gauss–Seidel iterative method is adopted. So, by applying Gauss–Seidel iterative method to Equations (18) and (19),

$$\overline{V}_{i+1}^{k+1} = V_i^{k+1} + chAV_{i+1}^k + h^2 A^2 V_i^{k+1}, \tag{20}$$

$$V_{i+1}^{k+1} = \frac{1}{4}\left(3V_i^{k+1} + \overline{V}_{i+1}^{k+1}\right) + h\left(aAV_i^{k+1} + bA\overline{V}_{i+1}^{k+1}\right), \tag{21}$$

where $k + 1$ is the current iteration number and $k$ is used for the previous iteration.

Since von Neumann stability analysis is a useful way to find the stability of the proposed scheme for linear or linearized partial differential equations, the von Neumann method has been applied to ordinary differential equations with the Gauss–Seidel iterative method in [25]. So, a similar approach will be employed in this contribution to find the

stability condition of Equations (20) and (21). To apply von Neumann stability analysis, consider the following transformations,

$$\left.\begin{array}{c} \overline{V}_{i+1}^{k+1} = \overline{E}^{i+1} e^{(i+1)I\psi}, \ V_i^{k+1} = E^{k+1} e^{iI\psi}, \\ V_{i+1}^{k} = E^k e^{(i+1)I\psi}, \ V_{i+1}^{k+1} = E^{k+1} e^{(i+1)I\psi}, \end{array}\right\} \tag{22}$$

where $I = \sqrt{-1}$.

Substituting some of the transformations from (22) into the first stage of the scheme (20) and simplifications yields,

$$\overline{E}^{k+1} = \left( I.De^{iI\psi} + h^2 A^2 \right) E^{k+1} + chAE^k, \tag{23}$$

where $I.D$ matrix denotes the identity matrix.

Now, substituting some transformations into the second stage of the scheme (21) and simplifications gives,

$$E^{k+1} e^{iI\psi} = \frac{1}{4} \left( 3E^{k+1} + \overline{E}^{k+1} e^{iI\psi} \right) + h \left( aAE^{k+1} + bA\overline{E}^{k+1} e^{iI\psi} \right). \tag{24}$$

Rewrite Equation (24) as,

$$\left[ I.D - \frac{3}{4} I.De^{-I\psi} - haA - \left( \frac{I.D}{4} + hbA \right) \left( I.De^{-I\psi} + h^2 A^2 \right) \right] E^{k+1} = \left( \frac{I.D}{4} + hbA \right) chAE^k. \tag{25}$$

The amplification factor is expressed as,

$$\left| \left( 1 - e^{-I\psi} - ah\lambda_A - \frac{h^2 \lambda_A^2}{4} - hb\lambda_A e^{-I\psi} - h^3 b\lambda_A^3 \right)^{-1} \left( \frac{1}{4} + hb\lambda_A \right) \right| \le 1, \tag{26}$$

where $\lambda_A$ is the maximum eigenvalue of matrix $A$.

Inequality (26) can be expressed as,

$$\left| \frac{1}{4} + hb\lambda_A \right| \le \left| 1 - e^{-I\psi} - ah\lambda_A - \frac{h^2 \lambda_A^2}{4} - hb\lambda_A e^{-I\psi} - h^3 b\lambda_A^3 \right|. \tag{27}$$

*2.2. Consistency Analysis*

For consistency analysis, Taylor series expansions will be adopted. Consider Equations (20) and (21) for the discretization of Equation (17).

At this stage of the procedure, substituting Equation (18) into Equation (19) yields,

$$V_{i+1} = \frac{1}{4} \left( 4V_i + chAV_{i+1} + h^2 A^2 V_i \right) + h \left( aAV_i + bAV_i + bchA^2 V_{i+1} + bh^2 A^3 V_i \right). \tag{28}$$

Rewrite Equation (28) as,

$$\left( I.D - \frac{1}{4} chA - bch^2 A^2 \right) V_{i+1} = \frac{1}{4} \left( 4I.D + h^2 A^2 V_i \right) + h \left( aAV_i + bAV_i + bh^2 A^3 V_i \right). \tag{29}$$

Consider Taylor series expansion for $V_{i+1}$ and substituting it into Equation (29) that results in,

$$\left( I.D - \frac{1}{4} chA + 0\left(h^2\right) \right) \left( V_i + hV_i' + 0\left(h^2\right) \right) = I.DV_i + hA(a+b)V_i + 0\left(h^2\right). \tag{30}$$

Rearranging Equation (30) yields,

$$V'_i = A\left(a + b + \frac{1}{4}c\right)V_i + 0(h).\tag{31}$$

Applying limit when $h \to 0$ in (31) gives,

$$V'_i = AV_i,\tag{32}$$

which is the original Equation (17) evaluated at grid point or time level "$i$".

### 3. SEIR Epidemic Model with Convex Incidence Rate

Let $S$ denote the susceptible people, which means people who have chances to become exposed. Exposed people are denoted by $E$, and these are those individuals that have germs of disease but do not show symptoms of the disease. The third category of the SEIR epidemic model is infected people. These are the people that have germ(s); they can spread the disease, and also symptoms of the disease are shown. This contribution suggests the contact rate of infected individuals having form $\beta SI(1 + \alpha I)$, where $\beta$ denotes the infection rate. This contact rate will denote the infection rate. This contact rate will rise or decay if the number of infected people increases or decreases respectively. The exposed people turn into infected individuals, and the corresponding time in which it occurs is denoted by $\frac{1}{v}$. Let $r$ represent the rate at which infected converts into the category of recovered people.

By following [26,27], the SEIR epidemic model with a convex incidence rate can be expressed as,

$$\frac{dS}{dt} = -\beta SI(1 + \alpha I) - \beta SE,\tag{33}$$

$$\frac{dE}{dt} = \beta SI(1 + \alpha I) + \beta SE - vE,\tag{34}$$

$$\frac{dI}{dt} = vE - \gamma I,\tag{35}$$

$$\frac{dR}{dt} = \gamma I,\tag{36}$$

subject to the initial conditions,

$$S(0) = S_0,\ E(0) = E_0,\ I(0) = I_0,\ R(0) = R_0,\tag{37}$$

where $S_0, E_0, I_0, R_0$ are constants.

*Equilibrium Points*

The disease-free equilibrium point can be found by solving,

$$0 = -\beta SI(1 + \alpha I) - \beta SI,$$
$$0 = \beta SI(1 + \alpha I) + \beta SE - vE,$$
$$0 = vE - \gamma I.$$

The disease-free equilibrium point can be expressed as,

$$B_0(S^*, 0, 0)$$

**Theorem 1.** *The system (33)–(36) is locally stable if it satisfies,*

$$v + \gamma - \beta > 0 \text{ and } \gamma v - \beta v - \beta \gamma > 0$$

**Proof .** To prove this theorem, the Jacobean eigenvalues will be determined. The Jacobean at the disease-free equilibrium point is given as,

$$J|_{B_0} = \begin{bmatrix} 0 & -\beta & -\beta \\ 0 & \beta - v & \beta \\ 0 & v & -\gamma \end{bmatrix} \tag{38}$$

The characteristics polynomial of the matrix (48) is given as,

$$p(\lambda) = \lambda^3 + (\gamma + v - \beta)\lambda^2 + (\gamma v - \beta \gamma - \beta v)\lambda \tag{39}$$

The characteristics equation is expressed as,

$$\lambda^2 + (\gamma + v - \beta)\lambda + (\gamma v - \beta \gamma - \beta v) = 0 \tag{40}$$

According to Routh-Hurwitz criteria for second-degree polynomials, the system will be stable, or both eigenvalues have a negative real part if,

$$v + \gamma - \beta > 0 \text{ and } \gamma v - \beta v - \beta \gamma > 0 \tag{41}$$

□

## 4. Results and Discussion

The suggested scheme is built with two grid points or time levels. This is one of the scheme's advantages because it does not require any additional initial conditions or the usage of any other scheme on the first grid point or time level. Schemes constructed on more than two grid points or time levels necessitate using another scheme or an initial guess to begin the solution method. However, any other scheme employs only the first time level or first grid point, and the scheme that is formed on two three-time levels or three grid points finds the solution at the remaining grid points. Because the first stage of the proposed scheme is implicit, an iterative strategy is used to solve the difference equation generated by the suggested scheme. A stopping criterion is addressed when finding a solution by coupling two methods. The stopping criterion is determined by the maximum of norms derived from two successive iterations. The iterative scheme's working technique depends on the initial guess, and the solution is discovered at the second grid point or time level utilizing the initial guess. So, the solution is found in the first iteration. For the rest of the iterations, the solution at the current iteration takes the information of the solution from the preceding grid point or time level.

The exact solution to the mathematical model under consideration is unknown. The highly accurate solution provided by the Matlab solver ode45 is used to test the scheme's performance. The Matlab solver ode45 can solve linear and nonlinear first-order ordinary differential equations. It necessitates the solution of a set of differential equations, a set of initial conditions, and the range of an independent variable in the form of a vector. It is possible that the Matlab solver will not converge in some case(s). However, in the present situation, the solver finds an accurate solution for chosen parameters. It is also possible that it will find a negative solution because it just solves provided differential equations and does not guarantee a positive solution. However, it can be utilized in instances where the solution is positive. The solution is also discovered using the first-order explicit Euler method and the nonstandard finite difference scheme (NSFD). The nonstandard scheme guarantees a positive solution and is also unconditionally stable. However, it has the disadvantage of not having even first-order accuracy. This disadvantage of the nonstandard finite difference approach for parabolic partial differential equations (PDEs) was highlighted in [28]. Furthermore, for parabolic PDEs, the nonstandard finite difference approach is conditionally consistent. The order of accuracy drawback of the nonstandard finite difference approach can be demonstrated using Taylor series expansions. This contribution presents a conditionally stable third-order approach. The proposed

technique is compared to the existing nonstandard finite difference method and the first-order Euler method. Figures 1–5 depict the absolute errors in susceptible, exposed, infective, and recovered individuals. The absolute error is determined by calculating the absolute difference between the proposed NSFD/Euler schemes and the Matlab built-in solver bvp4c. Table 1 compares the proposed scheme with existing numerical schemes for computing the maximum of norms and time consumed by each scheme. From Table 1, it can be seen that the proposed scheme produces less error than those given by existing schemes. The error is found by finding the maximum of all norms for all dependent variables in the considered SEIR model. This study also utilizes the neural network approach for mapping between input and targets in the form of susceptible, exposed, and infecteds. For fitting problems, a neural network is required for mapping between a set of inputs and a set of outputs. The examples include estimating house prices as targets and input variables, such as crime rate, pupil/teacher ratio in schools and tax rate and can estimate engine emission levels with the input of speed and fuel consumption. One more example can be body fat level related to the input of body measurements. Artificial neural network strategy has been given in [29] for the waterborne spread and control of diseases. The neural network fitting tool can evaluate performance using regression analysis and mean square error. If enough neurons are in the hidden layer of a two-layer feed-forward network with consistent data, then the network can be fitted to multi-dimensional mapping problems. The Levenberg–Marquardt back propagation algorithm will be employed for the network to be trained if there is enough memory; otherwise, scaled conjugate gradient backpropagation will be applied. Figures 6–8 show the mean square error over the iterations or epochs for susceptible, exposed, and infecteds, respectively. The mean square error decreases by increasing iterations or epochs. Figures 9–11 show the error histograms for susceptible, exposed, and infecteds, respectively. Figures 9–11 also show zero errors that represent the minimum all the errors. Figures 12–14 present the best lines for output data and targets.

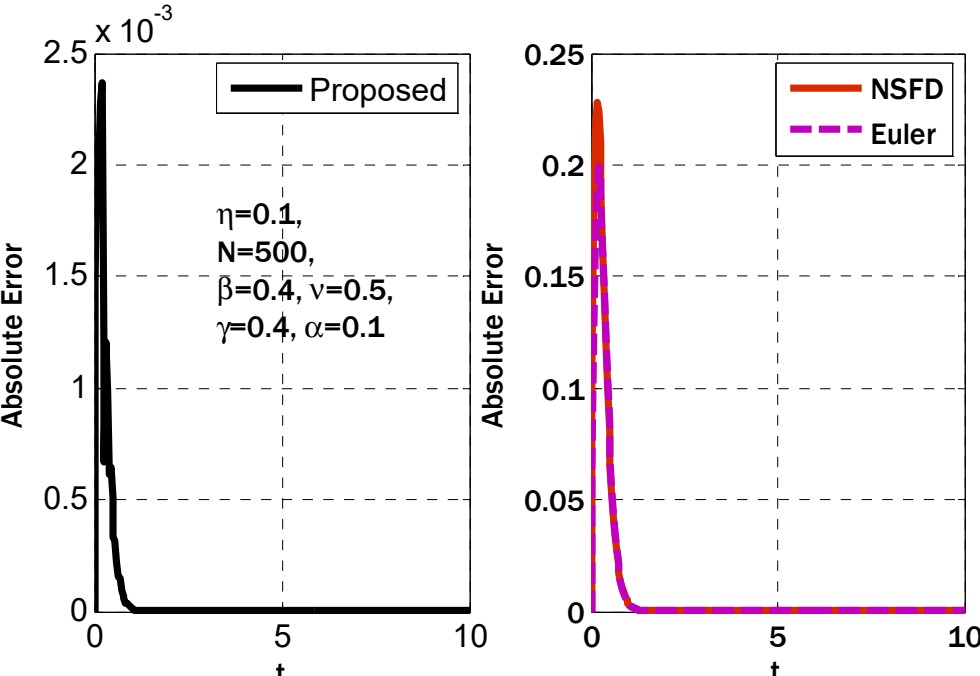

**Figure 1.** Comparison of three schemes for susceptible people using $\eta = 0.1$, $N = 500$, $\beta = 0.4$, $\nu = 0.5$, $\gamma = 0.4$, $\alpha = 0.1$.

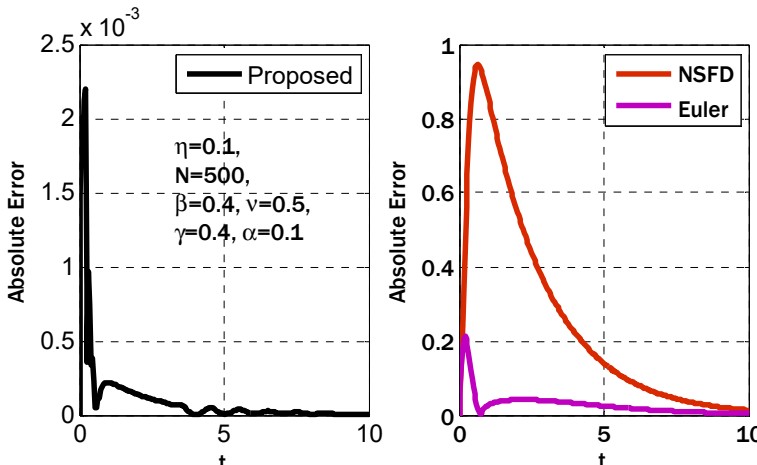

**Figure 2.** Comparison of three schemes for exposed people using $\eta = 0.1$, $N = 500$, $\beta = 0.4$, $\nu = 0.5$, $\gamma = 0.4$, $\alpha = 0.1$.

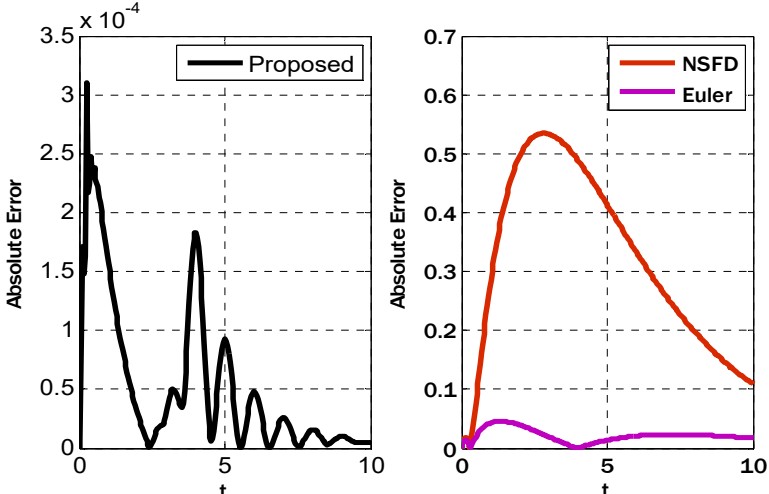

**Figure 3.** Comparison of three schemes for infected people using $\eta = 0.1$, $N = 500$, $\beta = 0.4$, $\nu = 0.5$, $\gamma = 0.4$, $\alpha = 0.1$.

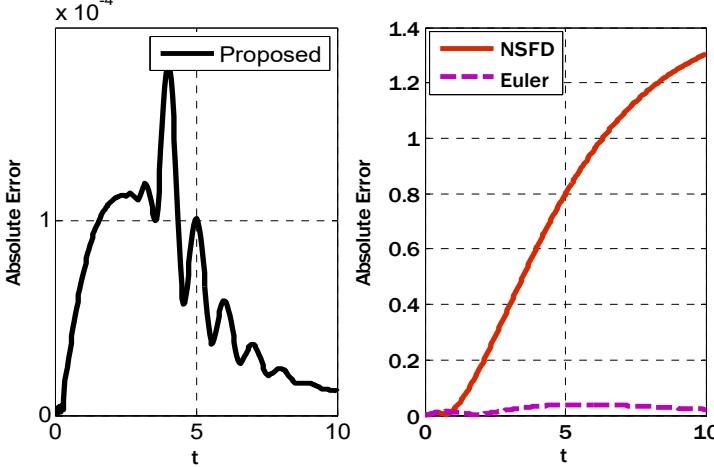

**Figure 4.** Comparison of three schemes for recovered people using $\eta = 0.1$, $N = 500$, $\beta = 0.4$, $\nu = 0.5$, $\gamma = 0.4$, $\alpha = 0.1$.

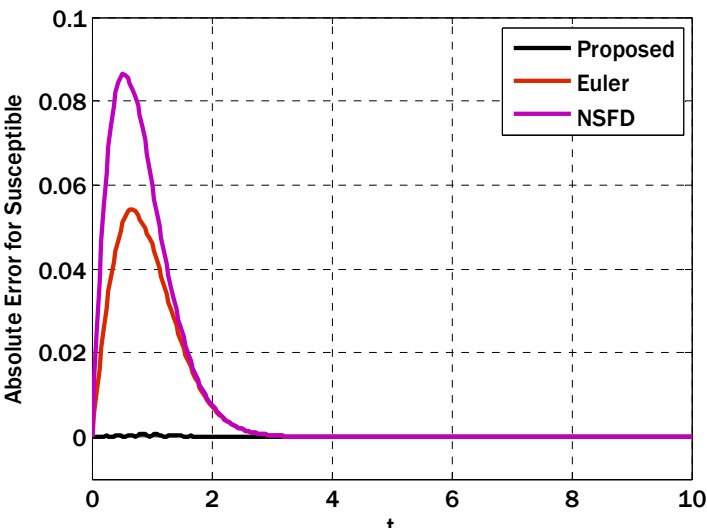

**Figure 5.** Comparison of three schemes for recovered people using $\eta = 0.3$, $N = 500$, $\beta = 0.1$, $\nu = 0.3$, $\gamma = 0.1$, $\alpha = 0.3$.

**Table 1.** Comparison of four schemes in finding the maximum of $L_2$ Norms and time consumed by each scheme.

| $N$ | Proposed Scheme | | NSFD | | Euler | | Adams-Bashforth | |
|---|---|---|---|---|---|---|---|---|
| | $L_2$ Norm | Time | $L_2$ Norm | Time | $L_2$ Norm | Time | $L_2$ Norm | Time |
| 25 | 0.0777 | 0.0155 | 11.0037 | 0.0114 | 1.4155 | 0.0121 | diverges | - |
| 50 | 0.0197 | 0.0257 | 7.8113 | 0.0128 | 0.8938 | 0.0121 | 0.2237 | 0.0148 |
| 100 | 0.0061 | 0.0303 | 5.5167 | 0.0167 | 0.6042 | 0.0154 | 0.0745 | 0.0151 |
| 200 | 0.0020 | 0.0480 | 3.8946 | 0.0176 | 0.4182 | 0.0163 | 0.0262 | 0.0174 |
| 400 | 0.0026 | 0.1025 | 2.7508 | 0.0219 | 0.2927 | 0.0217 | 0.0083 | 0.0170 |

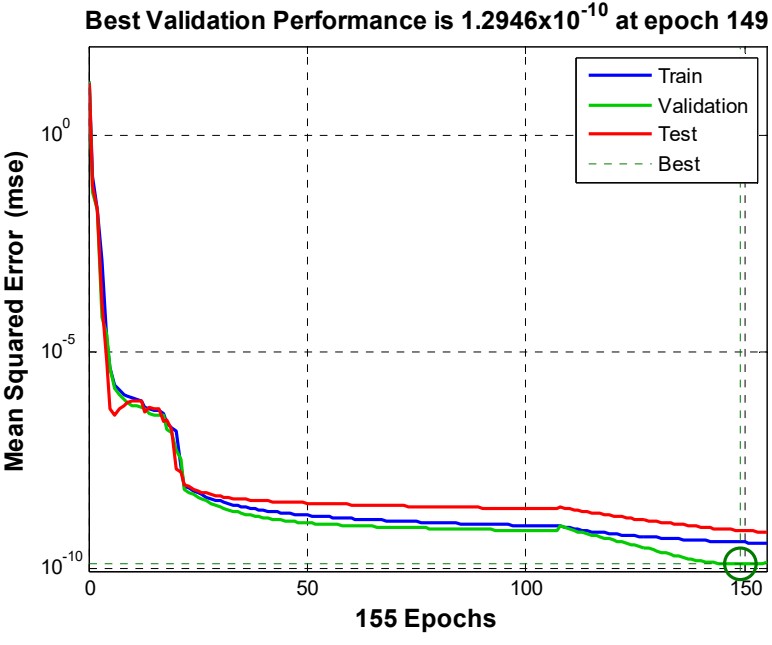

**Figure 6.** Mean square error over epochs for susceptible people using $\eta = 0.1$, $N = 500$, $\beta = 0.4$, $\nu = 0.5$, $\gamma = 0.4$, $\alpha = 0.1$.

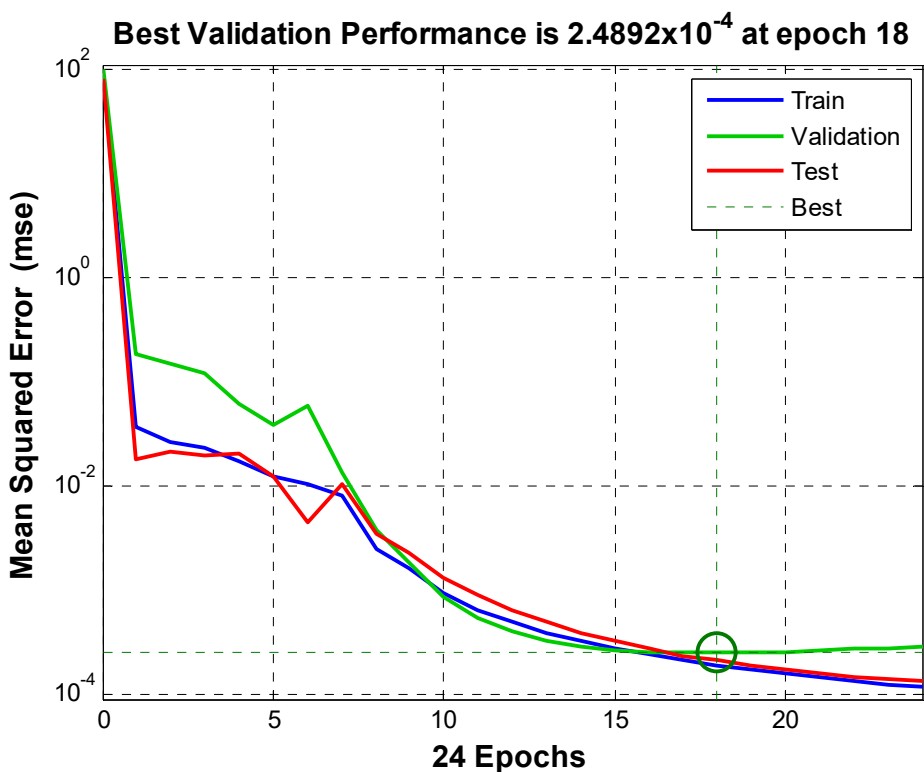

**Figure 7.** Mean square error over epochs for exposed people using $\eta = 0.1$, $N = 500$, $\beta = 0.4$, $\nu = 0.5$, $\gamma = 0.4$, $\alpha = 0.1$.

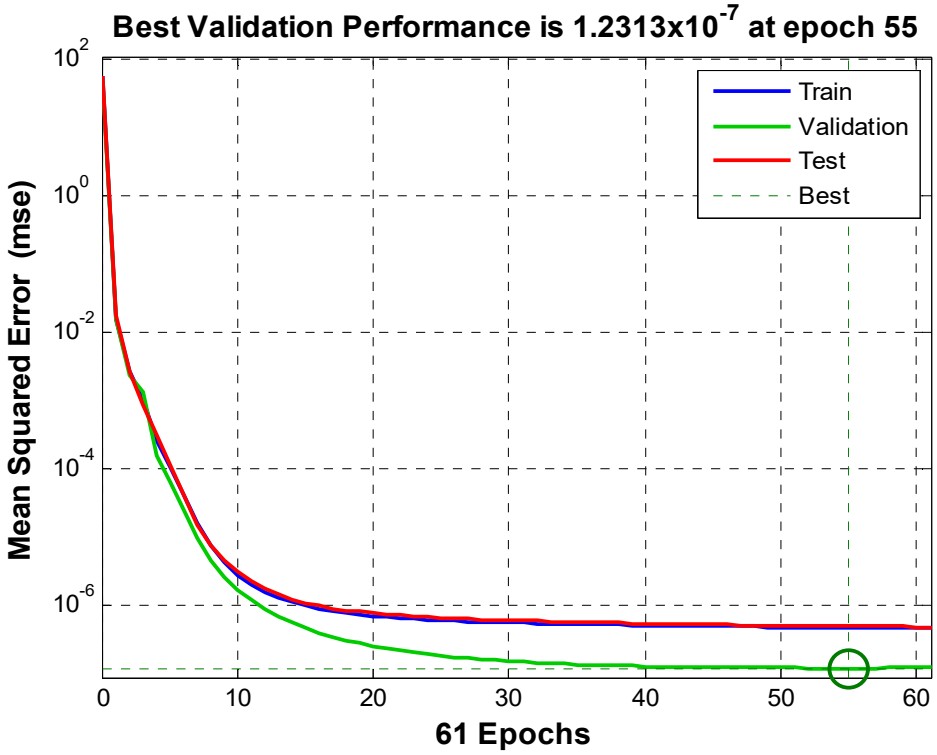

**Figure 8.** Mean square error over epochs for infected people using $\eta = 0.1$, $N = 500$, $\beta = 0.4$, $\nu = 0.5$, $\gamma = 0.4$, $\alpha = 0.1$.

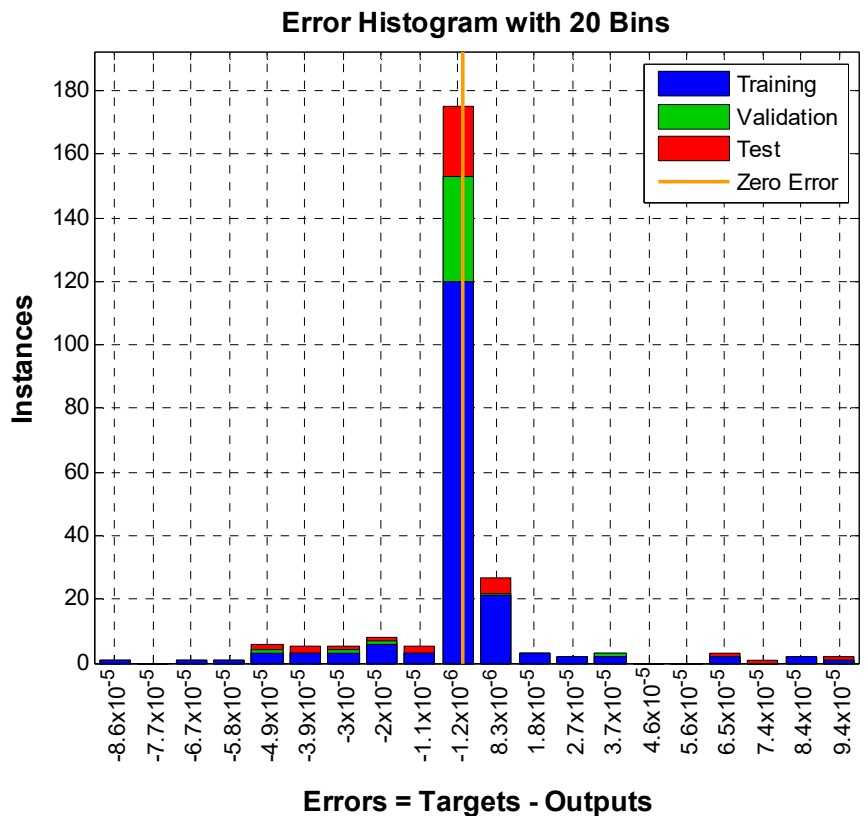

**Figure 9.** Error histogram for susceptible people using $\eta = 0.1$, $N = 500$, $\beta = 0.4$, $\nu = 0.5$, $\gamma = 0.4$, $\alpha = 0.1$.

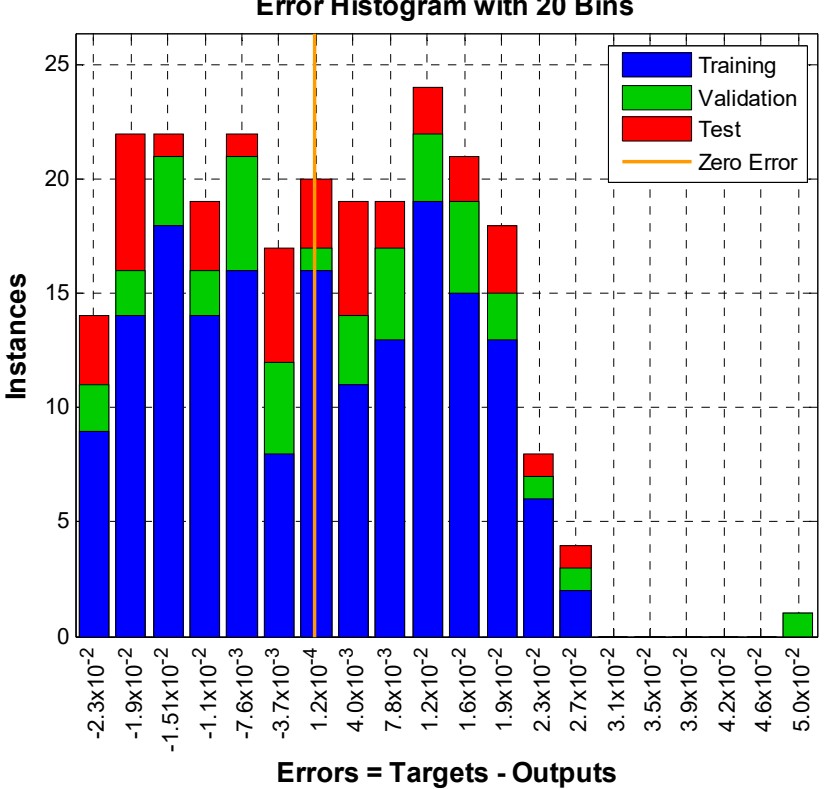

**Figure 10.** Error histogram for exposed people using $\eta = 0.1$, $N = 500$, $\beta = 0.4$, $\nu = 0.5$, $\gamma = 0.4$, $\alpha = 0.1$.

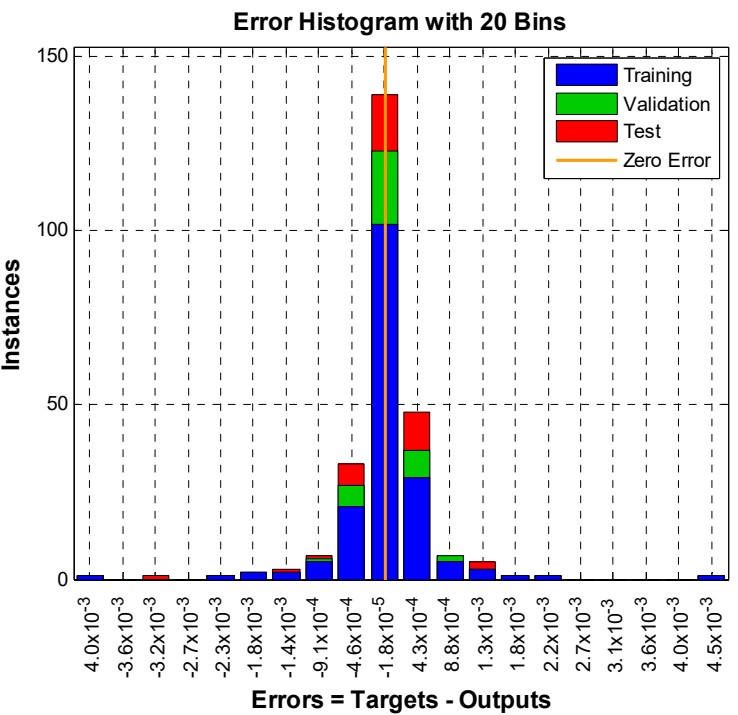

**Figure 11.** Error histogram for infective people using $\eta = 0.1$, $N = 500$, $\beta = 0.4$, $\nu = 0.5$, $\gamma = 0.4$, $\alpha = 0.1$.

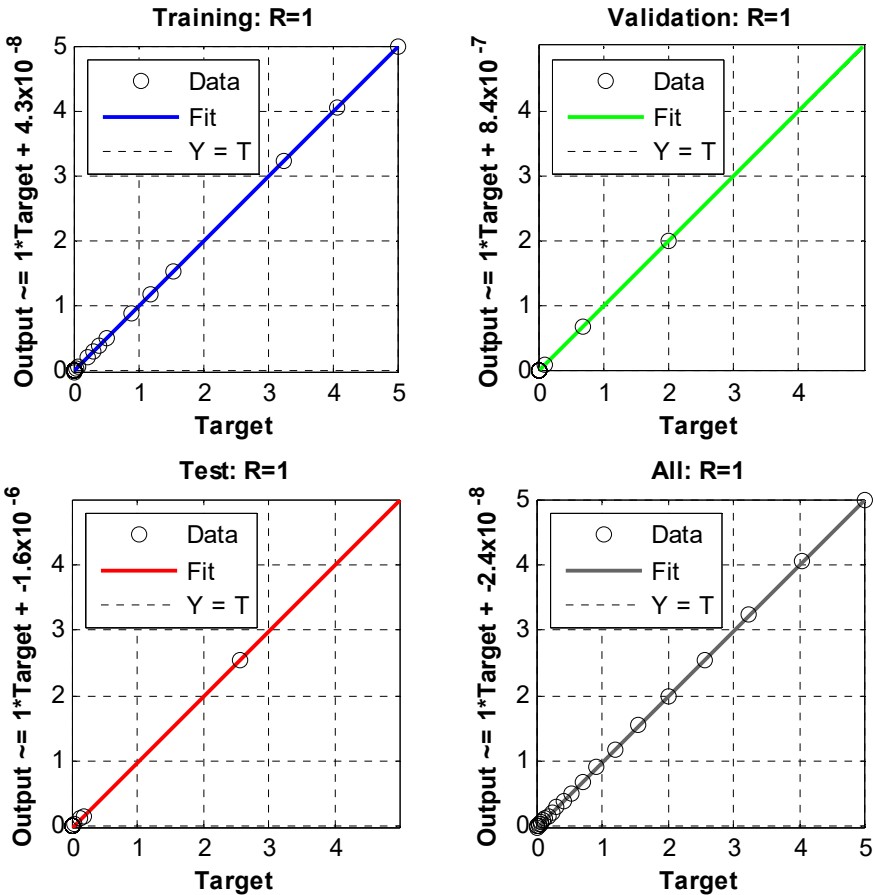

**Figure 12.** Lines of best fit for susceptible people using $\eta = 0.1$, $N = 500$, $\beta = 0.4$, $\nu = 0.5$, $\gamma = 0.4$, $\alpha = 0.1$.

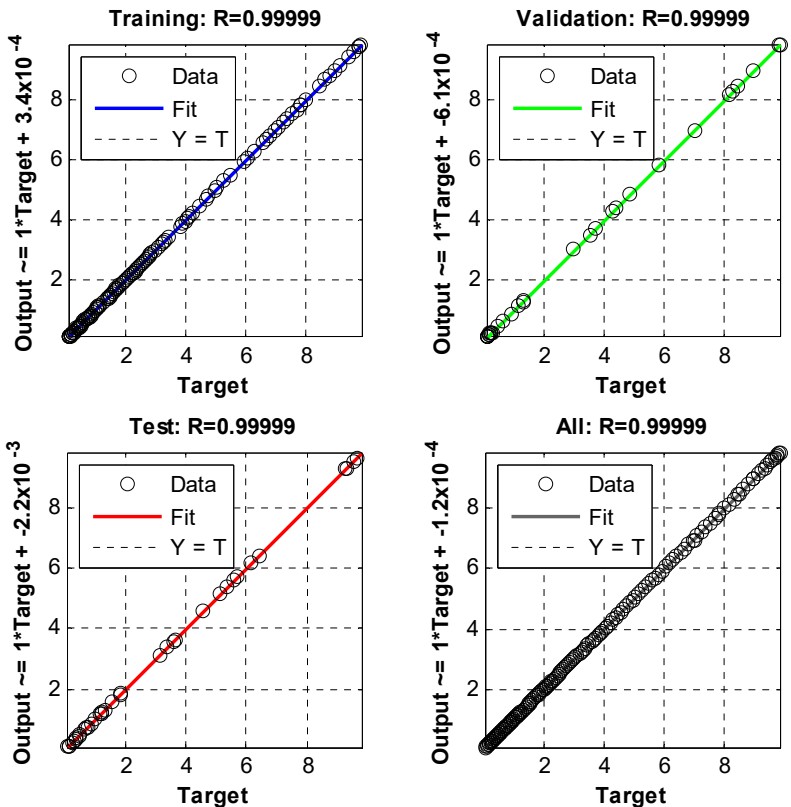

**Figure 13.** Lines of best fit for exposed people using $\eta = 0.1$, $N = 500$, $\beta = 0.4$, $\nu = 0.5$, $\gamma = 0.4$, $\alpha = 0.1$.

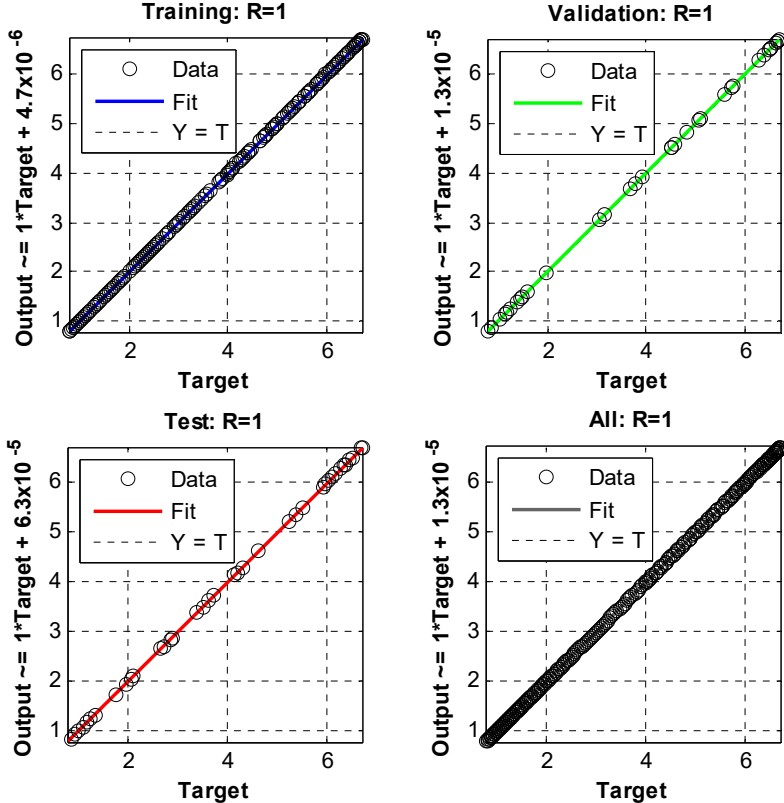

**Figure 14.** Lines of best fit for infective people using $\eta = 0.1$, $N = 500$, $\beta = 0.4$, $\nu = 0.5$, $\gamma = 0.4$, $\alpha = 0.1$.

## 5. Conclusions

A finite difference scheme has been proposed, which was an extension of the existing Runge–Kutta scheme. The scheme had one deficiency in that it required one extra derivative of the dependent variable. But it also provided third-order accuracy in two stages. The scheme has been successfully applied to solving nonlinear mathematical models of epidemic disease. Since the first stage of the scheme was implicit, an extra iterative procedure was adopted to solve difference equations, and these difference equations were obtained when the proposed scheme was applied to the SEIR model. Given some plots and tables of comparison, it could be deduced that the proposed scheme was better than the existing Euler method, and the existing Euler method was better than the nonstandard finite difference method in terms of accuracy for chosen specific parameter values.

**Author Contributions:** Methodology, Y.N.; Software, A.A.A.G., Y.N. and M.M.; Validation, H.J.A.S. and M.M.; Formal analysis, A.A.A.G., Y.N. and H.J.A.S.; Investigation, A.A.A.G.; Resources, Y.N. and H.J.A.S.; Writing—original draft, A.A.A.G.; Writing—review & editing, A.A.A.G., Y.N., H.J.A.S. and M.M.; Supervision, A.A.A.G. All authors have read and agreed to the published version of the manuscript.

**Funding:** This research was supported by Deanship of Scientific Research, Vice Presidency for Graduate studies and Scientific Research, King Faisal University, Saudi Arabia [grant No. 3794].

**Data Availability Statement:** Data is not available due to privacy.

**Conflicts of Interest:** The authors declare no conflict of interest.

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
