# Peer review of "Extended Runge-Kutta Scheme and Neural Network Approach for SEIR Epidemic Model with Convex Incidence Rate"

_processes, doi:10.3390/pr11092518_

Round 1

Reviewer 1 Report

Page 1 Paragraph 1: Please change/remove the sentence as it is abrupt and too general. Remove ";See" as the references are self-explanatory.

Page 1 Paragraph 2: Why is COVID not mentioned in a paper about epidemics. Please include relevant references to recent events in the introduction and provide some context as to how this can be of benefit in such situations.

Page 1 Paragraph 2: Include references to the different methods.

Page 2 Paragraph 1: Include references to SEIR model and the full form in the place where you mention the acronym first.

Page 2 Paragraph 1: Change "Tey" to "They". Include explanation of AB and the full form.

Page 2 Paragraph 3: More detailed explanation is needed on how many schemes are used to solve differential equations. Why is Runge Kutta selected to solve instead of any other methods? What are the schemes/developments that have been reported in particular to obtain solutions to models for pandemics? What are the limitations of these developments? After this, the authors should discuss their proposed scheme. The introduction should give readers best possible overview of the problem the author is trying to solve. It needs extensive revision.

Section 2: Implicit-Explicit Runge Kutta schemes are not new. There are many reports that have been already published. It is not clear whether the novelty of this work is in implementing this method for the SEIR model or in creating a general approach for solving models with differential equations. 

Section 2: Many of the equations can be moved to supplementary information especially those that are simply expansions or rearrangements of the previous equation. Current approach is more suited to a mathematics textbook with complete explanation of the working steps, rather than a research paper where key findings are preferred with working steps in supplementary information.

Section 3: It would be better if the authors considered a real life situation with an example of a disease and assumptions and used the model with some available data.

Line 282-295: Include references as to the source and size of the dataset and what are the exact variables.

Line 289: It is not clear why this algorithm is chosen over other algorithms for testing.

Overall Comment: The manuscript should give a clear overview of the main conclusions and improvements that have been proposed. The examples overall lack in details, and while the figures are detailed they are not adequately explained in the text. More references and real numbers are required, instead of only general equations and figures.

English language needs moderate revision, there are some typos (such as "Tey" instead of "They" mentioned above) and abrupt sentences. Also, extensive improvement is required in referencing.

Author Response

Reply to Reviewer 1

Page 1 Paragraph 1: Please change/remove the sentence as it is abrupt and too general. Remove ";See" as the references are self-explanatory.

Reply Some changes are made in the introduction part of revised version of the manuscript.

Page 1 Paragraph 2: Why is COVID not mentioned in a paper about epidemics. Please include relevant references to recent events in the introduction and provide some context as to how this can be of benefit in such situations.

Reply That whole sentence is removed now including references.

Page 1 Paragraph 2: Include references to the different methods.

Reply Some more references are provided and cited in the introduction.

Page 2 Paragraph 1: Include references to SEIR model and the full form in the place where you mention the acronym first.

Reply This change is made in revised version.\

Page 2 Paragraph 1: Change "Tey" to "They". Include explanation of AB and the full form.

Reply The correction is made now.

Page 2 Paragraph 3: More detailed explanation is needed on how many schemes are used to solve differential equations. Why is Runge Kutta selected to solve instead of any other methods? What are the schemes/developments that have been reported in particular to obtain solutions to models for pandemics? What are the limitations of these developments? After this, the authors should discuss their proposed scheme. The introduction should give readers best possible overview of the problem the author is trying to solve. It needs extensive revision.

Reply There exist many schemes in the literature for solving differential equations. Some of the schemes have been mentioned. Actually this work is based on the proposing or constructing the extension of existing Runge-Kutta method for solving differential equations. This scheme is new according to our knowledge, it does not exist in literature. So, this work is based on the proposing or constructing a numerical scheme with some of its analysis and its application to some mathematical model.

Section 2: Implicit-Explicit Runge Kutta schemes are not new. There are many reports that have been already published. It is not clear whether the novelty of this work is in implementing this method for the SEIR model or in creating a general approach for solving models with differential equations. 

Reply Implicit-explicit schemes are not new but this scheme is the extension of implicit-explicit scheme.

Section 2: Many of the equations can be moved to supplementary information especially those that are simply expansions or rearrangements of the previous equation. Current approach is more suited to a mathematics textbook with complete explanation of the working steps, rather than a research paper where key findings are preferred with working steps in supplementary information.

Reply Book contains existing work but this work is not based on the existing scheme but the book may contain more details. Details of steps are also given in the paper, it can be useful for wide readership but still some equations are removed from the manuscript.

Section 3: It would be better if the authors considered a real life situation with an example of a disease and assumptions and used the model with some available data.

Reply This is general model to test a scheme. Also, some effect was also incorporated with the SEIR model. The main of the work to propose a scheme.So, the scheme can be applied to any differential equation in science and engineering.

Line 282-295: Include references as to the source and size of the dataset and what are the exact variables.

Reply One reference is provided that also contains the study of neural network. The neural network is based on the Matlab built in utility. So, input and targets are provided and it can provide the results in form of figures.

Line 289: It is not clear why this algorithm is chosen over other algorithms for testing.

Reply Since, modification is made in the existing Runge-Kutta method so the new scheme is applied to the SEIR model along with comparison of some existing schemes.

Reviewer 2 Report

It is suggested that several sections for smoother English expression should be refined.

Author Response

Reply to Reviewer 2

  1. What is the connection between the final two sentences in the Abstract, and do they

constitute semantic repetition?

REply One sentence is removed from the abstract.

2 Some formats may need to be adjusted, for example,

(1)The curly brace utilized in equation (25) appears to encompass three lines, where-

as it only encompasses two lines in its current presentation. This inconsistency

requires justification;

Reply The change is made now.

(2)The manuscript exhibits an omission of periods within all equations and certain

references in last section of this manuscript. Kindly review is accordingly needed;

Reply Effort is put to make changes of commas and full stops.

(3)The caption under this figure and the number on X-label is too close and the

layout needs to be adjusted and optimized

Reply Figure is not mentioned. But still if journal has some problem related to any then some change or changes might have been made.

3 Some expressions may be inappropriate. Please pay particular attention to English

grammar, spelling, and sentence structure, so that the goals and results of the study

are clear to the reader. Take a small number of statements as an example:

  • Line 34, “aids” should be capitalized as ”AIDS”; while the word ”Where” in Line
    180 should use the lowercase;

Reply The whole sentence is removed with references.
(2)Line 38, “Epidemiologists utilize these plans” here used is not appropriate, which
should be modified to “Epidemiologists utilize these methods” or “Epidemiologists
utilize these tools”;

Reply Change is made.

(3)Line 39, the tense of this sentences should be checked, specifically, “ · · · ap-
proaches must be used to · · · ” should be changed to “ · · · approaches are used to
· · · ”;

Reply It is done.

(4)Line 53, “is made easier” should be “are made easier”;

Reply It is done.

(5)Line 58, “Tey” should be corrected to “They”;

Reply It is done.

(6)Line 81, “having” should be replaced with “with”;

Reply It is done.

(7)Line 82, the sentence “The disadvantage of the scheme is to find an extra deriva-
tive...” should be changed to “The disadvantage of the scheme is the need to calculate
an additional derivative...” to guarantee the fluency;

Reply It is done.

(8)Line 103, “The symbol “h” is Equation (3) · · · ” should be changed to “The
symbol ”h” in Equation (3) · · · ”;

Reply The effort is put to make some change.

(8)Line 137, the expression “· · · it can linearized · · · ” should rephrased using the
passive voice;

Reply It is done.

(9)Line 327, the sentence ”The has been · · · ” is incorrect and unclear. Please review
and revise it for clarity;

Reply It is corrected now.

  1. The language in certain parts of the paper appears to be a direct translation from y-

our native language, which occasionally affects the flow and coherence of the text. To

enhance readability and engagement, consider refining these sections for smoother

English expression. Additionally, infusing your writing with a more refined style

will not only enhance clarity but also elevate the impact of your research.

Reply Some changes have been made.

Reviewer 3 Report

This works deals with the extension of the Runge-Kutta scheme and Neural Network approach to solve numerically a SEIR epidemic model with convex incidence rate.

I read the manuscript with a great attention and think that it is well presented and the results seem correct. Nevertheless, there are some  points which must be addressed before its acceptance for publication.
1- The main contribution must be highlight in the Introduction section used bullets;
2- The organization of the manuscript is missed;
3- Several statement (Line 65-79) without any references? Which references support these affirmations?
4- The authors must use the punctuation sign after each equation. For example, there are coma after equations (1) and (2), and full stop after equation (3)
5- The authors must illustrate in the same Figures, the proposed method and the NSFD and Euler. For each scheme, the accuracy and time-convergence must be given in a Table ( for example).

Author Response

Reply to Reviewer 3

  • The main contribution must be highlight in the Introduction section used bullets;

Reply The introduction in extended is revised form of manuscript.

  • The organization of the manuscript is missed;

Reply The organization of the manuscript is added in the introduction.

  • Several statement (Line 65-79) without any references? Which references support these affirmations?

Reply This paragraph is related to schemes but some references related to methods has been added before in the introduction.

  • The authors must use the punctuation sign after each equation. For example, there are coma after equations (1) and (2), and full stop after equation (3)

Reply Effort is put in these changes.

  • The authors must illustrate in the same Figures, the proposed method and the NSFD and Euler. For each scheme, the accuracy and time-convergence must be given in a Table ( for example).

Reply One figure is provided for comparison of solutions in one plot and one table is also provided for comparison of different schemes.

Reviewer 4 Report

I read this paper in detail. Authors need to revise it via follows:

1- [1- 5] these papers should be written in detail.

2- [9,10]=> [9, 10]

3- [17-21] these papers are also the same.

4- Equations (1) and (2) should be ended with comma.

5- Equations (3) should be ended with point.
6-Author should review the whole manuscript and correct all the grammatical and typo
errors.

7- Authors should pay attention to punctuation marks at the end of equations.

8-Equation (10) should be written in properly.

9-Equations (40-43) should be cited by at least one related paper.

10-They need to compare their model via existing models such as "Heuristic computing with sequential quadratic programming for solving a nonlinear hepatitis B virus model; An Efficient Numerical Scheme for Biological Models in the frame of Bernoulli Wavelets; The dynamical analysis of a Tumor Growth model under the effect of fractal fractional Caputo-Fabrizio derivative; Fractional SIZR model of Zombies infection".  What is the main contrbiution and advantage of their model shuld be explained in detail.

11- Neural method need to be also compared via "Levenberg-Marquardt back propagation neural network procedures for the consumption of hard water-based kidney function". This scheme must be compard and explained in clear.

After these modifications, it may be accepted.

I read this paper in detail. Authors need to revise it via follows:

1- [1- 5] these papers should be written in detail.

2- [9,10]=> [9, 10]

3- [17-21] these papers are also the same.

4- Equations (1) and (2) should be ended with comma.

5- Equations (3) should be ended with point.
6-Author should review the whole manuscript and correct all the grammatical and typo
errors.

7- Authors should pay attention to punctuation marks at the end of equations.

8-Equation (10) should be written in properly.

9-Equations (40-43) should be cited by at least one related paper.

10-They need to compare their model via existing models such as "Heuristic computing with sequential quadratic programming for solving a nonlinear hepatitis B virus model; An Efficient Numerical Scheme for Biological Models in the frame of Bernoulli Wavelets; The dynamical analysis of a Tumor Growth model under the effect of fractal fractional Caputo-Fabrizio derivative; Fractional SIZR model of Zombies infection".  What is the main contrbiution and advantage of their model shuld be explained in detail.

11- Neural method need to be also compared via "Levenberg-Marquardt back propagation neural network procedures for the consumption of hard water-based kidney function". This scheme must be compard and explained in clear.

After these modifications, it may be accepted.

Author Response

Reply to Reviewer 4

  • [1- 5] these papers should be written in detail.

Reply Some detail is provided for these references.

  • [9,10]=> [9, 10]

Reply These reference including text are removed.

  • [17-21] these papers are also the same.

Reply Some detail is also added for these references.

  • Equations (1) and (2) should be ended with comma.

Reply It is done now.

  • Equations (3) should be ended with point.

Reply It is done.

  • Author should review the whole manuscript and correct all the grammatical and typo

Reply Some changes are made.

  • Authors should pay attention to punctuation marks at the end of equations.

Reply Some/most/all changes regarding this problem are made now.

  • Equation (10) should be written in properly.

Reply It is approximately re-arranged now.

  • Equations (40-43) should be cited by at least one related paper.

Reply Two references are provided for the model with the considered effect.

  • They need to compare their model via existing models such as "Heuristic computing with sequential quadratic programming for solving a nonlinear hepatitis B virus model; An Efficient Numerical Scheme for Biological Models in the frame of Bernoulli Wavelets; The dynamical analysis of a Tumor Growth model under the effect of fractal fractional Caputo-Fabrizio derivative; Fractional SIZR model of Zombies infection".  What is the main contrbiution and advantage of their model shuld be explained in detail.

Reply In the mentioned papers, one of the existing schemes was considered. So that existing scheme is also considered in this study for comparison purpose.

11- Neural method need to be also compared via "Levenberg-Marquardt back propagation neural network procedures for the consumption of hard water-based kidney function". This scheme must be compard and explained in clear.

Reply These figures of neural network are obtained from the Matlab built-in utility. So these figures also discuss about the errors in the outcome and targets.Levenberg-Marquardt back propagation is also one of the algorithms chosen by the built-in utility. 

Round 2

Reviewer 1 Report

1. Line 23-24 " that includes [1] 23 which deals with COVID-19 epidemic model that considers the effect of lock-down and" its important that authors write in scientific language. For example line can be modified as "that includes author name et al year  [1] 23 which deals". Please correct throughout out the manuscript.

1. Line 23-24 " that includes [1] 23 which deals with COVID-19 epidemic model that considers the effect of lock-down and" its important that authors write in scientific language. For example line can be modified as "that includes author name et al year  [1] 23 which deals". Please correct throughout out the manuscript.

Reviewer 3 Report

This version of the manuscript is well improved. The authors have addressed all Reviewer's comments. I recommend the manuscript for publication.